# Prediction method for the reaction force of vehicle door sealing systems

**Zhong Yang**[1,2], **Luoxing Li**[1,3]*, **Jing Huang**[2], **Guoqing Chen**[4], **Zhengqing Liu**[4], **Zhenhu Wang**[5]

**1** State Key Laboratory of Advanced Design and Manufacturing for Vehicle Body, College of Mechanical and Vehicle Engineering, Hunan University, Changsha, Hunan, China, **2** Deepal Automobile Technology Co., Ltd, Chongqing, China, **3** Research Institute of Hunan University in Chongqing, Hunan University, Chongqing, China, **4** College of Mechanical Engineering, Zhejiang University of Technology, Hangzhou, Zhejiang, China, **5** College of Mechanical Engineering, Hunan Institute of Engineering, Xiangtan, Hunan, China

* luoxing_li@hnu.edu.cn

## Abstract

The accuracy of calculating the reaction force in vehicle sealing systems using moment equilibrium equations from statics is highly dependent on the degree of simplification in the mathematical model. This study demonstrates that employing the Finite Element (FE) method allows for a comprehensive consideration of the coupling effects between the door and the flexible seal, thereby eliminating errors inherent to model simplification and single-boundary variables. Consequently, this approach improves the prediction accuracy of the sealing system's reaction force. In this work, a multi-boundary coupling simulation method for a vehicle door sealing system was established using the Finite Element (FE) method, which accounts for the interaction between the seal and the complex door structure. The results, based on data from three distinct vehicle models (with three doors tested per model), show a mean absolute percentage error (MAPE) of 7% between the simulated and experimentally measured static closing forces. This close agreement verifies the reliability of the proposed method. This study provides a comprehensive strategy for predicting and optimizing the reaction force in vehicle sealing systems. It was found that a 75% increase in the compression load deflection (CLD) of the lock-side seal yields a 22% increase in the total sealing system reaction force.

## 1. Introduction

The rapid electrification of the automotive industry has heightened consumer expectations for a refined driving experience, placing greater emphasis on vehicle Noise, Vibration, and Harshness (NVH) performance. Previous studies [1–3] have established that the sound quality of door opening and closing is a critical factor in user perception and product satisfaction, serving as a key indicator of overall vehicle NVH

**Data availability statement:** All data are in the manuscript and supporting information files.

**Funding:** The authors would like to thank the National Natural Science Foundation of China (Grant number 5247121335), the Hunan Provincial Department of Education Youth Project (Grant number 22B0742) and the National Natural Science Foundation of China (Grant number 5227120232) for their financial support for this research. However, the funders had no role in study design, data collection and analysis, decision to publish, or preparation of the manuscript.

**Competing interests:** The authors declare that they have no known competing financial interests or personal relationships that could have appeared to influence the work reported in this paper. The authors declare the following financial interests/personal relationships which may be considered as potential competing interests.

performance. The reaction force of the sealing system is a vital metric for this sound quality, directly influencing its characteristics. Insufficient sealing system reaction force can compromise the seal between the door and the body frame, leading to door vibration and dynamic sealing issues [4]. Conversely, an excessively large reaction force not only increases the required door-closing effort and amplifies the unlocking sound, thereby reducing acoustic comfort, but also imposes excessive mechanical stress on the lock system [5–9]. Consequently, the accurate prediction and control of the sealing system's reaction force have become a pressing task for automotive engineers seeking to maintain a competitive edge through enhanced user perception and satisfaction. This study aims to systematically analyze the reaction force of a New Energy Vehicle (NEV) door sealing system using simulation methods and to propose innovative optimization strategies to improve door opening and closing sound quality. Wagner et al. [10] pioneered the nonlinear analysis of seal compression load, while Wang et al. [11] developed an Excel-based mathematical model to calculate door-closing force. However, these methods essentially simplify the door and sealing system into a single, decoupled model, neglecting their interaction and consequently introducing errors into the results.

To address the aforementioned factors, this study proposes a multi-boundary coupling simulation method for the vehicle door sealing system. This approach utilizes finite element analysis to construct an integrated door-seal model and maps the seal's Compression Load Deflection (CLD) curve onto the full-vehicle FE model. The coupled model captures the deformation of both the door and the seal under their flexible interaction, enabling an accurate simulation of the seal reaction force during door-closing events. Furthermore, based on a contribution analysis, the study implements a systematic optimization of the seal's CLD values. This optimization provides a foundation for controlling the reaction force and guiding the seal's structural design. To validate the method's effectiveness and general applicability, tests were conducted on three representative vehicle models.

## 2. Numerical simulation

### 2.1. Theoretical

As shown in Fig 1, static door closure force is defined as the minimum external force perpendicularly applied to the door outer panel at the position corresponding to the latch engagement point, required to achieve complete door closure prior to the initiation of latch engagement [12–13]. The door seal can be decomposed into several consecutive segments. Under the condition that door deformation and seal compression deformation are not considered, the total torque acting on the seal is given as follows:

$$E_{\mathrm{md}} = \sum_{i=1}^{N} F_i L_i \tag{1}$$

where $E_{md}$ denotes the total torque, $I$ represent the total number of segments of the seal, $F_i$ is the force applied to the $i$-th segment of the seal, $L_i$ stands for the moment arm from the centroid position of the $i$-th segment of the seal to the hinge axis.

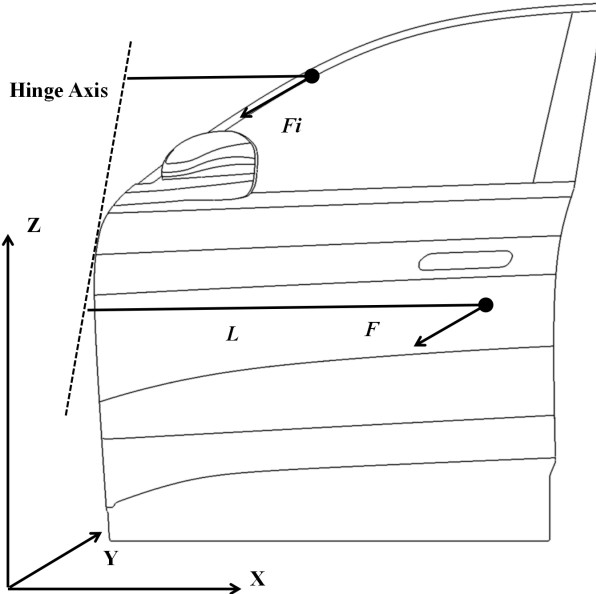

**Fig 1. Schematic diagram of vehicle door force distribution.**

According to the principle of torque equilibrium, the force at any arbitrary point on the vehicle door with respect to the hinge axis is expressed as follows:

$$F = E_{\mathrm{md}}/L_s = \frac{(\sum_{i=1}^{N} F_i L_i)}{L_s} \qquad (2)$$

where $L_s$ denotes the moment arm from any arbitrary point on the door to the door hinge axis. The static closing force of the door is primarily contributed by three components: the door seal, latch mechanism, and buffer blocks. Among these, the latter two account for only 20%–25% of the total static reaction force. This study focuses on analyzing the reaction force distribution within the sealing system.

## 2.2. Vehicle door sealing system

The structural configuration of the vehicle door sealing system is schematically illustrated in Fig 2(a). This integrated system employs a hierarchical sealing architecture, consisting of the door seal (primary sealing element) and door frame seal (secondary sealing element), which serve as the primary and secondary sealing interfaces to establish hermetic seals between adjacent door panels [12–14]. The dynamic sealing performance is further augmented by the glass run channel and clamping strips, which provide dynamic sealing interfaces during window actuation. Considering that the interference between the door seal and the vehicle body varies at different positions, while the interference between the door frame seal and the door remains consistent, adjustments are made to unify the interference parameters of the door frame seal with those of the door seal. Both the door frame and door seals are classified into four distinct functional zones: the door lock position, hinge position, sill position, and header position, as shown in Fig 2(b) and Fig 3. This zoning strategy takes into account the differences in compression amount and compression direction of the seal at different positions. Such an analysis facilitates targeted optimization of the seal.

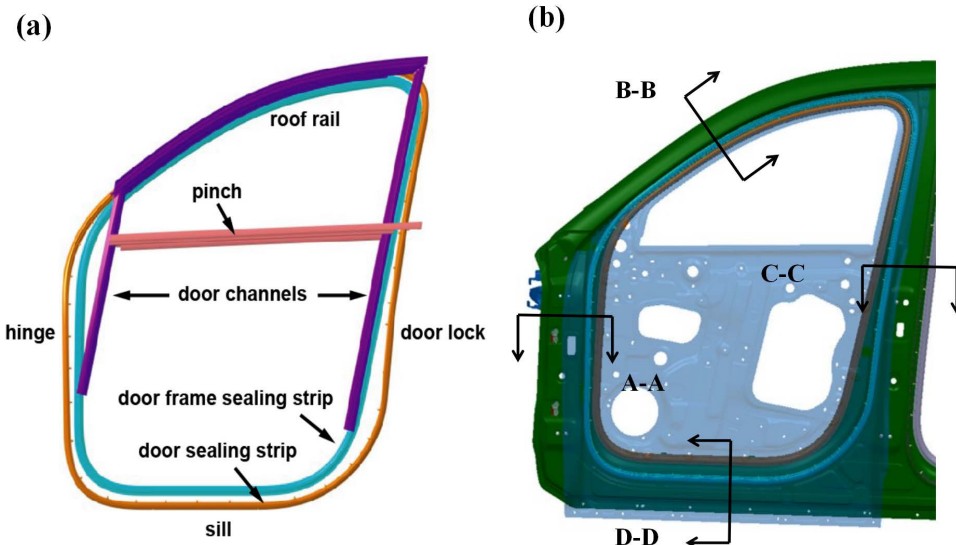

**Fig 2. Schematic diagram of (a) vehicle door sealing system and (b) sealing strip.**

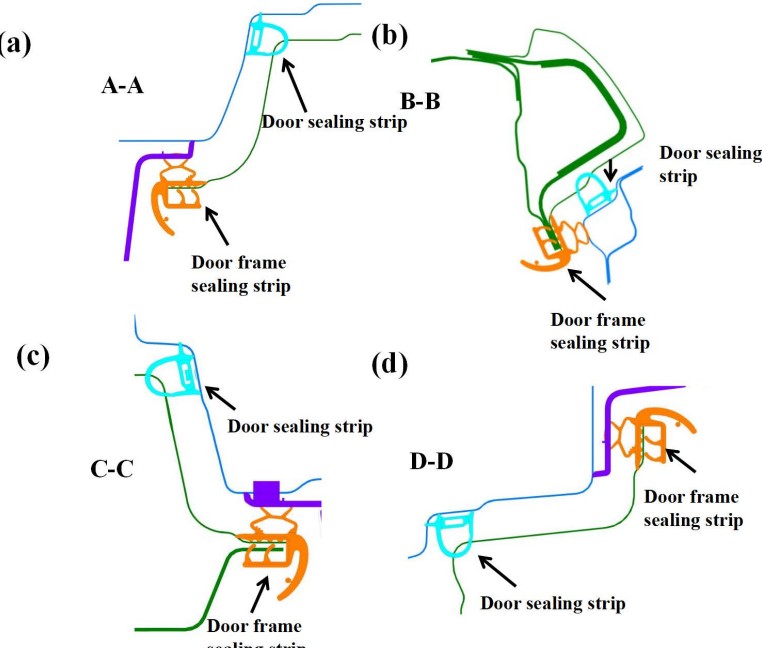

**Fig 3. Section view of the vehicle door sealing system corresponding to Figure 2(b).**

## 2.3. Finite element modeling

Taking the front door sealing system of a specific SUV model as a case study, this paper develops a multi-boundary coupled simulation model for the door-seal system. The model is used to analyze the reaction forces generated by seal compression under static conditions, extract the reaction force at the door handle, and quantify the contribution of reaction

forces across different seal segments. These results provide a theoretical basis for optimizing CLD values in each seal segment.

The 3D door geometry was imported into finite element preprocessing software, where redundant features were removed and the model simplified prior to geometric input. Given that the thickness dimension of door panels is significantly smaller than the other two dimensions, 2D shell elements were generated using mid-surface extraction. Mesh quality assurance is critical for reliable analysis results. During modeling, key structural regions were preserved while simplifying non-critical components to balance computational accuracy and efficiency through controlled node density. The average mesh size was set to 8 mm, with element types including TRIA3 and QUAD4.

After the modeling process is completed, the components of the car door system were assembled and connected via bolts, hinges, and seals (as shown in Fig 4). Bolts were simulated by means of RBE2 elements. Hinges were modeled in such a way that the rotational degrees of freedom were released using RBE2 elements. Seals were modeled through RBE3 + CBUSH + RBE3 connections. Each elastic element was spaced at intervals of 100 mm to simulate the seal stiffness per 100 mm. Subsequently, the seal's CLD curve is mapped to the CBUSH elements. This study primarily focuses on the seal reaction force exerted on the door at the moment of equilibrium, which is a static equilibrium process. The internal deformation, friction, and air resistance of the seal are not investigated herein.

Following assembly, material properties and thickness values were assigned to the finite element model in accordance with the bill of materials (BOM). Spot welds, seam welds, and plug welds were modeled based on welding process data. The final finite element model is presented in Fig 5. Mesh characteristics are summarized as shown in Table 1 and Table 2, and the FE modeling workflow is shown in Fig 6.

## 2.4. D. segment calibration for CLD

The CLD of a seal, i.e., compression load-deflection, describes the relationship between the reaction force generated by the seal when compressed and the amount of compression. $D0$ refers to the compression amount of the door compressing the seal under the design condition, with the unit of mm. The CLD curve characterizes the mechanical behavior of seal strips during compression. Through this curve, the compression performance and load-bearing capacity of seal strips can be systematically evaluated, which is critical for selecting appropriate materials, structural configurations, and vulcanization degrees. During the seal design process, CLD curves serve as essential references. By adjusting material properties,

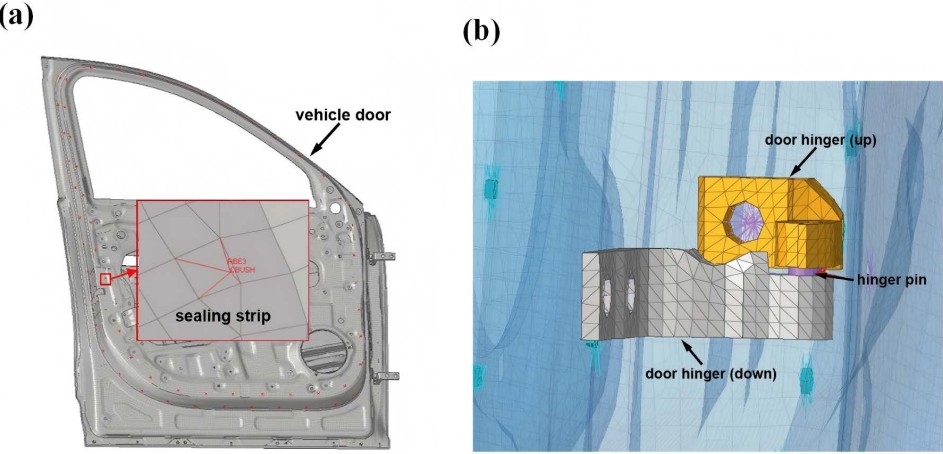

**Fig 4. Coupled FE model of (a) sealing strip and (b) hinge.**

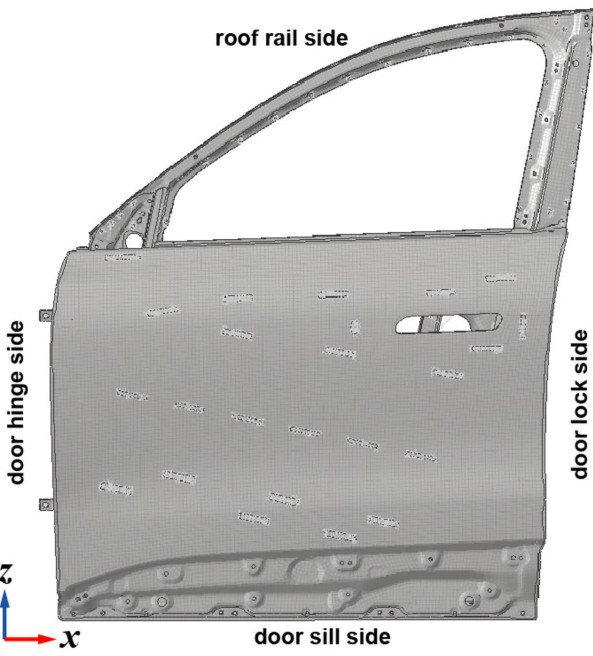

**Fig 5. Multi-boundary FE model.**

**Table 1. Mesh statistics.**

| Total Weight | Number of Elements | Number of Nodes | Triangular Elements |
| --- | --- | --- | --- |
| 25.8 kg | 85755 | 81852 | 4.8% |

**Table 2. Material parameters.**

| Material | Elastic Modulus (MPa) | Poisson's Ratio | Density (KG/m³) |
| --- | --- | --- | --- |
| Steel | 206000 | 0.3 | 7830 |
| Spot Weld | 206000 | 0.3 | 7830 |
| Expansion glue | 1.63 | 0.41 | 1400 |
| Structure glue | 1515 | 0.41 | 1400 |
| Glass | 72000 | 0.28 | 2200 |
| PVC | 100 | 0.4 | 1400 |

geometric parameters, and vulcanization levels, CLD values under different compression states can be optimized to meet application-specific requirements. Rational CLD values are critical for balancing reaction forces in sealing systems.

Method One offers the advantage of predicting CLD values at the conceptual design stage, thereby guiding reaction force calculations and CLD parameter optimization. However, its accuracy relies on detailed material property inputs to ensure constitutive model fidelity. Conversely, Method Two provides high precision and reliability. To ensure the accuracy of the research method, Method Two was adopted in this study to determine the CLD values. Compression load tests were conducted on the door seals and door frame seals. The test equipment and scenarios are illustrated in Fig 7.

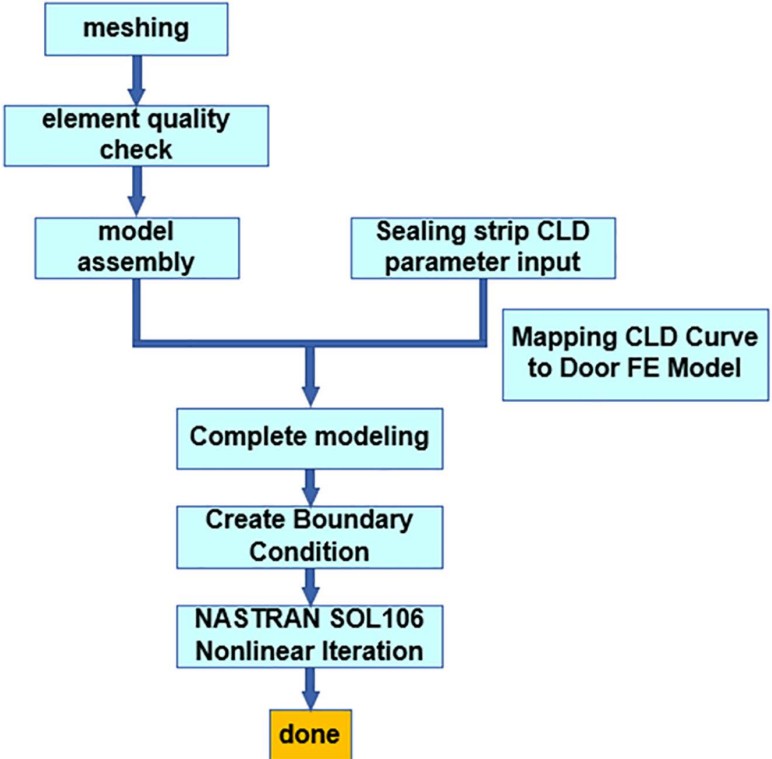

**Fig 6. Analysis flowchart.**

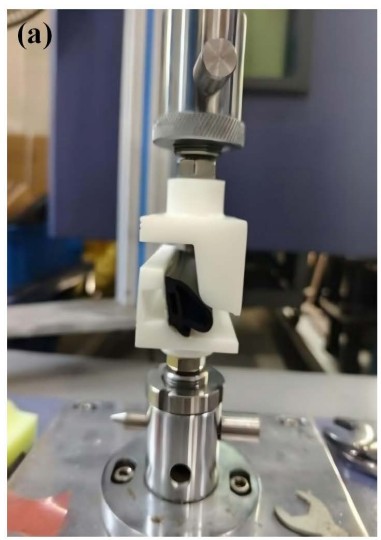 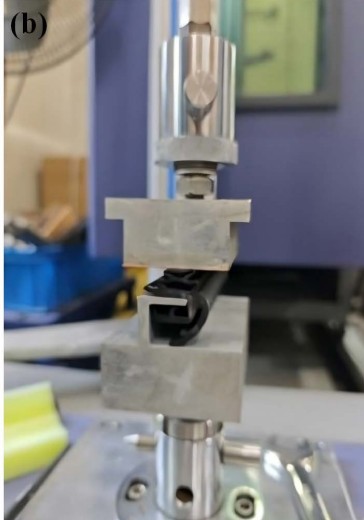

**Fig 7. Test Bench for seal compression load testing.**

At room temperature (23 ± 2) °C, three seal samples (each with a length of 100 ± 1 mm) were obtained from the test positions along the length of the seal (see Fig 8). The selection of positions refers to Fig 2 (b). The test samples were mounted on the tooling bench and placed at the center of the lower platform to ensure no pre-compression; the upper pressure plate was adjusted until it lightly touched the surface of the sample (with a contact force ≤ 0.1 N), and the displacement sensor was reset to zero. The sample was compressed at a speed of 30 mm/min until the target deformation was reached, and this target deformation was maintained for 10 seconds. Finally, each sample was tested three times, and the average value was taken. The results for each position were obtained in the same manner, and the test results were compiled into a report. The key parameters of the seal strip CLD are presented in Table 3.

Force-displacement curves of the seal strips were obtained via cross-sectional simulations, as shown in Fig 9 and Fig 10. Furthermore, the test results were used as input data for the analysis of the sealing system's reaction force.

## 3. Simulation analysis

The CLD parameters of the seal were mapped into the door FE model, and boundary conditions were applied to the model. All degrees of freedom (123456) of the mounting points at the joints between the upper and lower hinges and the vehicle body were constrained, while allowing the hinges to rotate along the hinge axis. A Y-direction constraint was applied at the observation point (actual measurement point). The Nastran SOL106 nonlinear solver was used to simulate the force condition of the seal during compression when the door is in the closed state, thereby analyzing the reaction force of the sealing system.

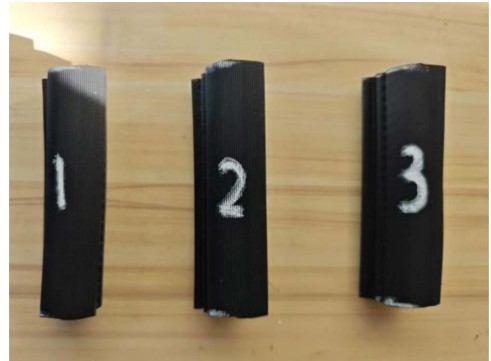

**Fig 8. Schematic diagram of seal test specimens.**

**Table 3. CLD Parameters of Door Frame Seal and Door Seal.**

| Position | | D0-2 N/100 mm | D0 N/100 mm | D0 + 2 N/100 mm |
|---|---|---|---|---|
| Door Strip N/100 mm | Hinge | 4.5 | 5.42 | 6.13 |
| | Roof Rail | 5.2 | 6.6 | 8.81 |
| | Door Lock | 4.82 | 5.58 | 6.02 |
| | Sill | 5.59 | 7.31 | 9.46 |
| Frame Strip N/100 mm | Hinge | 0.83 | 4.37 | 7.09 |
| | Roof Rail | 0.81 | 4.36 | 6.91 |
| | Door Lock | 0.69 | 4.38 | 7.21 |
| | Sill | 0.76 | 4.46 | 7.12 |

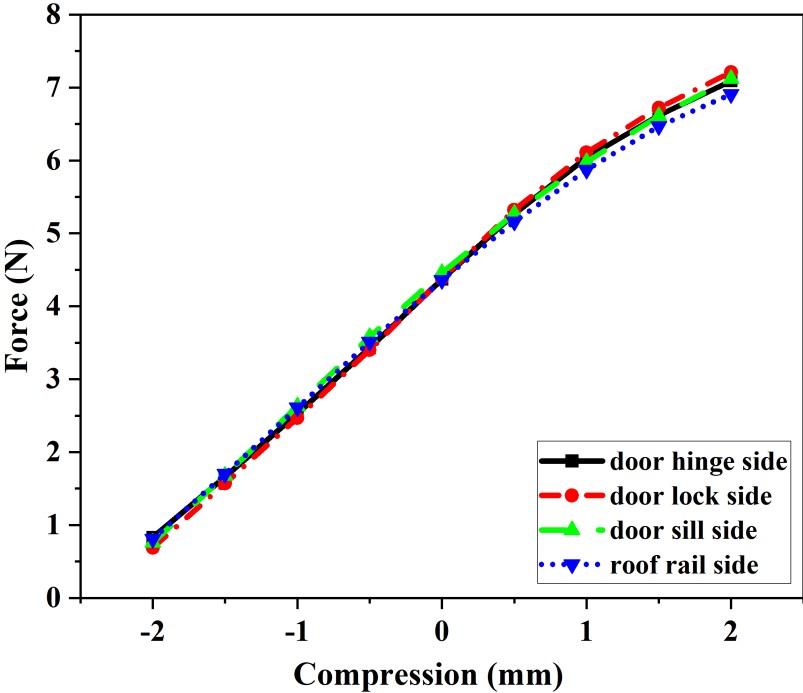

**Fig. 9. CLD Curve of Door Frame Seal.**

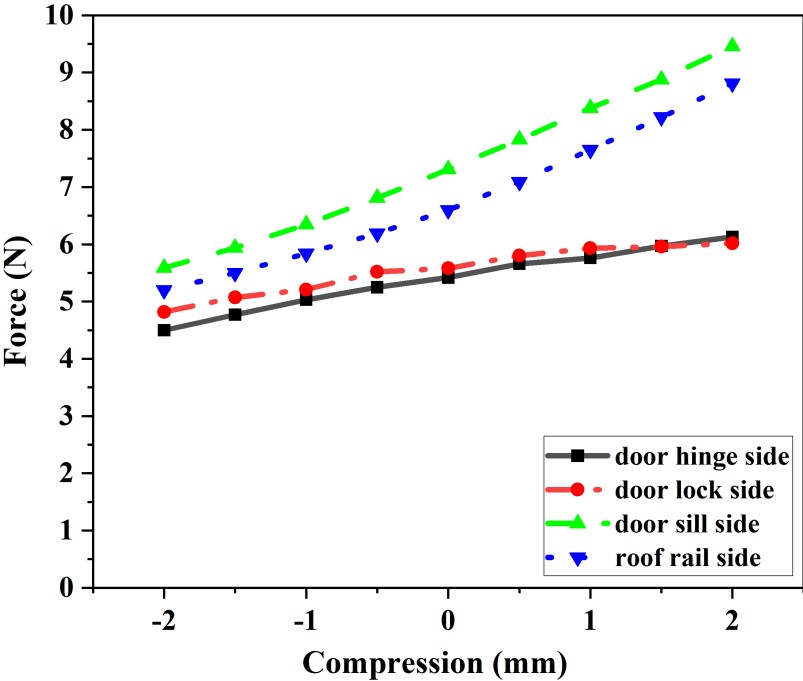

**Fig. 10. CLD Curve of Door Seal.**

## 3.1. Simulation results

The observation point is located 30 mm from the door edge on the outer panel corresponding to the door lock (consistent with the actual measurement point). The reaction force magnitude at the observation points and the displacement of the door frame in the Y-direction were output. The stress nephogram and displacement nephogram are shown in Figs 11 and 12, respectively.

This study identified a reaction force magnitude of 173 N at measurement point through nonlinear finite element analysis, which served as the basis for parametric optimization of CLD characteristics in the seal strip. The optimization process systematically adjusted seal geometry and material properties to achieve target CLD values, ensuring optimal force distribution across the sealing interface. Concurrent computational analysis evaluated Y-direction deformation of the door frame under the measured reaction force, yielding critical insights into structural response patterns. These deformation results were directly incorporated into the digital modeling workflow to inform pre-deformation compensation strategies for the door structure. The proposed approach leveraged nonlinear computational methods to account for bidirectional deformation interactions between the seal strip and door system. This modeling strategy demonstrated improved fidelity in predicting real-world structural responses compared to traditional linear methods. The simulation fully accounts for the flexibility of the door and its interaction with the seal, reducing the errors associated with the linear calculation method for reaction forces.

## 3.2. Contribution analysis

It can be derived from Formula 2 that the magnitude of the reaction force of the sealing system at the observation point (test point) is equal to the sum of the torques of the four regions divided by the moment arm from the observation point to the hinge axis. When the door structure is fixed, the moment arm becomes a constant value. The contribution analysis

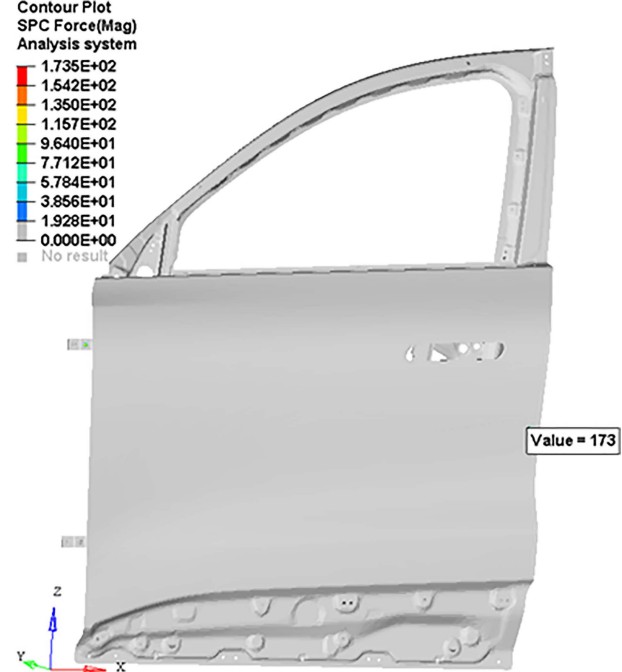

**Fig 11. Reaction force contour at the observation point.**

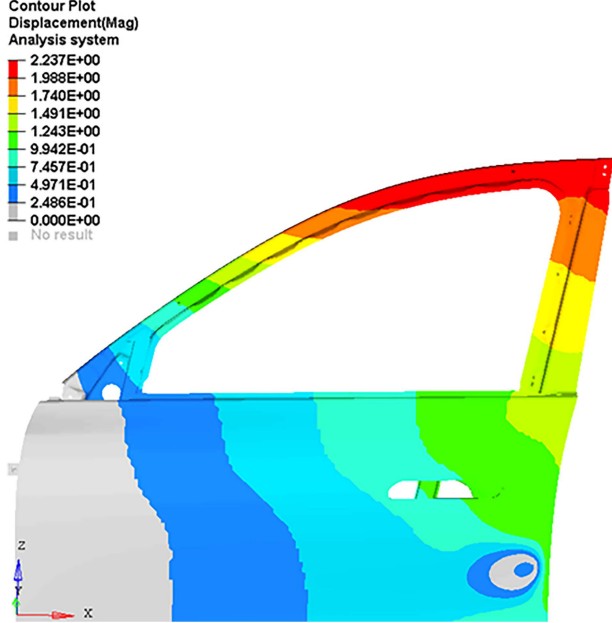

**Fig 12. Y-direction displacement contour of the door frame.**

is conducted under the assumption that the door structure is fixed and the moment arm remains unchanged, primarily to identify the influence of the seal's CLD on the reaction force.

To evaluate the contribution of the seal's CLD in a single region to the reaction force, the seals in the door lock, hinge, threshold, and crosshead regions were respectively removed from the finite element model to calculate the reaction force. Other boundary conditions were kept unchanged, and a single-factor variable method was adopted to compare and analyze the changes in reaction force before and after removing the seal in a specific region, thereby evaluating the reaction force contribution of the removed seal in that region.

To assess the influence of the segmented positions of the sealing strips on the reaction force, the reaction forces of the front – door sealing systems of three distinct models were analyzed independently. The reaction – force values at different positions of the various sealing strips are presented in Table 4 and Table 5.

Through the analysis of the reaction force contributions of the sealing systems in three different car doors, it can be observed that the seal strip areas each make distinct contributions to the reaction force of the sealing system. The reaction force contributions vary across different positions. Specifically, the contribution from the door – lock side of the seal strip is greater than that from other areas. This is because the lever arm from the door lock to the hinge is longer. The contribution from the hinge side is the smallest, while the contributions from the header side and sill side are moderate. However, there is a slight difference in the contribution ratio between frameless doors and framed doors; meanwhile, the

**Table 4. Contribution of different regions of the door seal to the sealing system reaction force (N).**

| Vehicle Type | Door Lock Side (N) | Contribution rate (%) | Roof Rail Side (N) | Contribution rate (%) | Hinge Side (N) | Contribution rate (%) | Sill Side (N) | Contribution rate (%) |
|---|---|---|---|---|---|---|---|---|
| Sedan A | 24 | 18% | / | / | 2 | 1% | 17 | 13% |
| SUV B | 45 | 26% | / | / | 2 | 1% | 27 | 15% |
| SUV C | 50 | 29% | 21 | 12% | 2 | 1% | 25 | 14% |

Table 5. Contribution of different regions of the door frame seal to the sealing system reaction force (N).

| Vehicle Type | Door Lock Side (N) | Contribution rate (%) | Roof Rail Side (N) | Contribution rate (%) | Hinge Side (N) | Contribution rate (%) | Sill Side (N) | Contribution rate (%) |
|---|---|---|---|---|---|---|---|---|
| Sedan A | 28 | 21% | 9 | 7% | 3 | 2% | 17 | 13% |
| SUV B | 39 | 22% | 9 | 5% | 2 | 1% | 16 | 9% |
| SUV C | 18 | 10% | 6 | 3% | 1 | 1% | 12 | 7% |

magnitude of the contribution ratio is also related to the value of the CLD. In the early stages of seal strip design, the CLD values for different areas can be designed according to their respective contributions to the reaction force. This approach can meet the varying reaction force requirements of different areas of the seal strip.

When the reaction force of the door sealing system needs to be optimized and increased, the total reaction force can be improved by enhancing the CLD curve of the seal on the door lock side. When the CLD force value of the door frame seal on the door lock side under the D0 condition increases from 4.38 N to 7.67 N, the reaction force of the sealing system increases from 173 N to 212 N. Specifically, the CLD curve of the door lock-side seal is enhanced by 75%, and the total reaction force of the sealing system increases by 22%.

## 4. Vehicle verification

To verify the reliability and broad applicability of this analytical method, simulations and tests of the sealing system reaction force were conducted on three different vehicle models, including those with framed doors and frameless doors. The simulation analysis method was consistent with that described in Chapter 3.1. The test method was aligned with the definition of static closing force, and the test results were compared with the simulation results.

The test equipment and scenarios are illustrated in Fig 13. Tests were performed on the prototype vehicles, ensuring that the state of the prototype vehicles was consistent with the designed state. The door was adjusted to the closed position, while the door lock remained in the unlocked state. A force gauge (with an accuracy of ±1 N) was used to slowly apply a Y-direction thrust at the position of the outer panel corresponding to the door lock (30 mm from the edge of the outer panel) until the door was locked. The reading of the force gauge at this moment was recorded as the door closing force. The average value of three test results was taken as the final test result and documented. A comparison between the simulation results and test results of the sealing system reaction force is presented in Table 6.

The measurement uncertainty is shown in Fig 14.

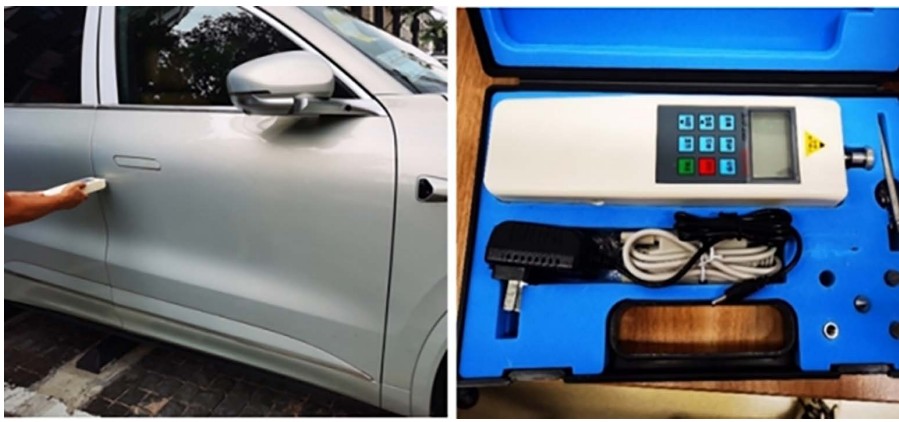

**Fig 13. Static closing force test.**

**Table 6. Simulation and test results of sealing system reaction force.**

| Vehicle Model | Average Test Value (N) | Sealing System Reaction Force Test Value (N) | Sealing System Reaction Force Test Value (N) | Absolute Percentage Deviation APE | Mean Absolute Deviation MAE(N) | Mean Absolute Deviation MAPE | Root Mean Square Deviation RMSE (N) |
|---|---|---|---|---|---|---|---|
| Sedan A | 213±1.4 | 150 | 136 | 10% | 14 | 3% | 14 |
| SUV B | 256±6.6 | 193 | 188 | 2% | 7 | 2% | 7.2 |
| SUV C | 249±2.2 | 186 | 173 | 7% | 13 | 7% | 13 |

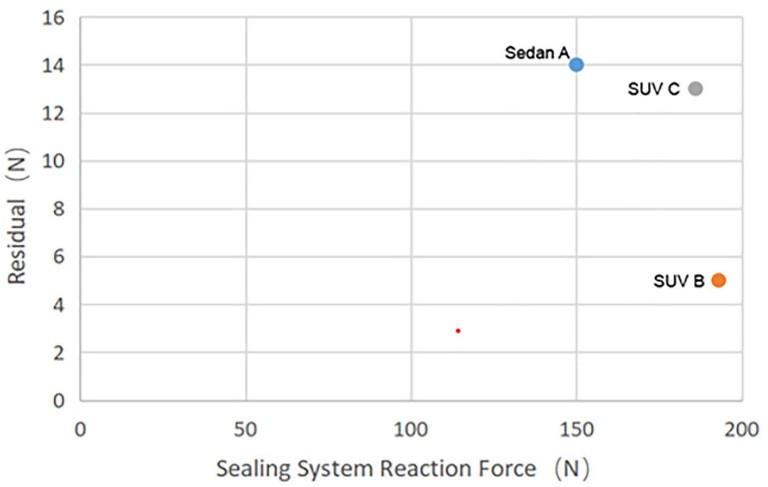

**Fig 14. Sealing system reaction force.**

Sealing system reaction force tests were conducted on three vehicle models. The static door closing force consists of the sealing system reaction force, the reaction force from the door lock, and that from the buffer block. The total reaction force provided by the door lock and buffer block is approximately 63 N; therefore, the reaction force provided by the sealing system can be calculated as shown in Table 6. The mean absolute percentage error (MAPE) between the sealing system reaction force simulation results and the experimental results is 6.3%, with a maximum error of 10%. A residual plot was generated based on the experimental results, which shows that the residuals fluctuate around zero. This confirms the consistency of data trends and indicates that the analysis results are reliable. Thus, the prediction method for the door sealing system reaction force established in this study provides reliable methodological support for the design and optimization of seals in the early stage of project development.

## Conclusion

This study demonstrates that the proposed simulation method for the sealing system reaction force effectively resolves the inaccuracies inherent in simplified Excel-based mathematical models. A key advantage of this method is its incorporation of the coupling effect between the door and the flexible seal, as well as the impact of the seal's post-compression deformation on the reaction force.

Using the front door of a specific SUV as a case study, we established an analytical framework for the door sealing system reaction force. The method was further validated through a comparative analysis of simulation and experimental results across three distinct vehicle models. The results indicate a mean absolute percentage error (MAPE) of 6.3%

between the simulated and measured static closing forces, with a maximum error of 10% across the three doors per model (n = 3). This close agreement verifies the method's effectiveness and reliability.

Furthermore, a contribution analysis of the sealing system reaction force quantified the influence of different seal regions. These findings informed a segmented optimization strategy for the seal's design. Additionally, data from the three vehicle models confirm that this simulation method is applicable to both framed and frameless door designs. By enabling accurate prediction and control of the reaction force to guide seal design according to project-specific requirements, the method proves to be both portable and promotable.

Nevertheless, a limitation of this study is that the simulation method is currently only suitable for predicting the sealing system reaction force in the final, closed state of the door. It does not model the dynamic door-closing process or the associated energy expenditure. Simulating these dynamic conditions is more complex, and methods for doing so remain underdeveloped, representing a key direction for future research.

## Author contributions

**Conceptualization:** Luoxing Li.

**Data curation:** Zhong Yang.

**Formal analysis:** Jing Huang.

**Investigation:** Guoqing Chen.

**Resources:** Zhengqing Liu, Zhenhu Wang.

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
