## [Decision Letter · Decision Letter 0]

3 Sep 2025

Dear Dr. Li,

Thank you for submitting your manuscript to PLOS ONE. After careful consideration, we feel that it has merit but does not fully meet PLOS ONE’s publication criteria as it currently stands. Therefore, we invite you to submit a revised version of the manuscript that addresses the points raised during the review process.

We look forward to receiving your revised manuscript.

Kind regards,

Cebastien Joel Guembou Shouop, Ph.D., ME, MS

Academic Editor

PLOS ONE

Journal Requirements:

2. Thank you for stating the following financial disclosure: [This work was funded by the National Natural Science Foundation of China (Grant number 52472378), Hunan Provincial Department of Education Youth Project (Grant number 22B0742), and the National Natural Science Foundation of China (Grant number 52272362)]. 

3. Please update your submission to use the PLOS LaTeX template. The template and more information on our requirements for LaTeX submissions can be found at http://journals.plos.org/plosone/s/latex .

5. Please provide a complete Data Availability Statement in the submission form, ensuring you include all necessary access information or a reason for why you are unable to make your data freely accessible. If your research concerns only data provided within your submission, please write "All data are in the manuscript and/or supporting information files" as your Data Availability Statement.

6. Thank you for stating the following in the Acknowledgments Section of your manuscript: [This work was funded by the National Natural Science Foundation of China (Grant

3 number 5247121335), the Hunan Provincial Department of Education Youth Project (Grant

4 number 22B0742) and the National Natural Science Foundation of China (Grant number

5 5227120232).]

Please remove any funding-related text from the manuscript and let us know how you would like to update your Funding Statement. Currently, your Funding Statement reads as follows: This work was funded by the National Natural Science Foundation of China (Grant number 52472378), Hunan Provincial Department of Education Youth Project (Grant number 22B0742), and the National Natural Science Foundation of China (Grant number 52272362)]. 

7. Please amend either the title on the online submission form (via Edit Submission) or the title in the manuscript so that they are identical.

8. Please amend either the abstract on the online submission form (via Edit Submission) or the abstract in the manuscript so that they are identical.

Additional Editor Comments:

Dear Dr. Luoxing Li,

Thank you for submitting your manuscript to PLOS ONE. After editorial assessment and external peer review, we find that your work addresses an important topic in vehicle sealing system modeling and validation. However, before we can proceed further in the evaluation process, the manuscript requires a major revision to meet the standards of clarity, methodological transparency, and reproducibility expected at PLOS ONE.

Below, we provide detailed feedback that we ask you to address in your revision.

**Major revisions**

1. Scope and positioning (NEV acoustics vs. static closing force).

The Introduction devotes substantial space to NEV sound quality and motor noise papers, but the study actually measures and simulates static reaction force for door sealing. Tighten the scope to the sealing system and static closing force; keep NVH framing only insofar as static force affects slam/noise. Remove or compress content on DC-motor noise models unless it directly informs your sealing model or validation.

2. Method/model coherence (hyperelastic seals vs. CBUSH springs).

You state a nonlinear FE approach accounting for hyperelasticity and bidirectional door–seal coupling; however, the implementation describes RBE3 + CBUSH elements spaced every 100 mm to represent seal stiffness (lumped springs). That is inconsistent with the earlier claim of fully coupled hyperelastic contact. Either (a) upgrade the seal to a continuum (solid) hyperelastic model with contact, or (b) explicitly present a reduced-order spring model, explaining how CLD curves are mapped to CBUSH nonlinear force–deflection properties and why this is acceptable. Right now, it reads contradictory.

3. Equations & units (Eqs. 1–3).

Clarify every symbol and unit. In equation 1, each term is not defined, and the paragraph that follows describes something else, which makes the reading difficult to follow. The terms are Fj, Fmf, Fs, and Fh in the equation, whereas in the following description, Fm, Fx, and d0 are described. It is unrelated. The definition “F = Fm/100” is unexplained (per-100-mm normalization?) and will confuse readers. Provide a consistent dimensional analysis (N per 100 mm vs. N total), define L, Ls, Ex (E?), and re-derive Eq. (3) cleanly. State all assumptions (rigid door? small deflection? linearization?) for the baseline equilibrium method you aim to supersede.

4. Boundary conditions, contact, solver settings.

For reproducibility, specify: contact type (surface-to-surface penalty? augmented Lagrange?), friction coefficient(s), damping, load steps/increments, convergence criteria, and solver (implicit/explicit). Define the applied load location/patch and how “pre-compression/interference” of the seal is realized. Present a mesh-independence and time-step sensitivity (if explicit) or load-step sensitivity (if implicit).

5. Validation design & error analysis.

Table 7 mixes multiple “test values” and a single error column; text says “average of three trials,” but the table shows three numbers (min/avg/max?) without labels. Define the statistic you compare (mean ± SD) and the exact error metric (absolute %, MAE, MAPE, RMSE). Provide measurement uncertainty (gauge ±1 N vs. operator variability) and show a Bland-Altman or simple residual plot to demonstrate agreement. The “≤10%” claim should be supported with a clear protocol and n for each vehicle.

6. Contribution (CLD) analysis method.

Explain precisely how you decomposed the regional contributions (door lock, hinge, sill, header). If you scaled regional CLDs and recomputed reaction forces, describe the parametric protocol (one-at-a-time vs. factorial) and how lever-arm effects were separated from local CLD changes. Right now, Tables 5–6 provide numbers, but the algorithm is not reproducible.

7. Figures & tables (labels, units, readability).

Add units to all color bars; ensure axis labels specify direction and units (e.g., “Displacement Y (mm)”). Correct formatting issues (e.g., bullet points in tables, column headings). Include sample sizes and error bars where relevant.

8. Language, typos, and technical terms.

Fix typographic/terminology issues (e.g., RBE2 not “BRE2”; “deformation,” not “eformation”). Streamline phrasing and avoid superlatives (“significant advancement”) unless quantitatively justified.

9. PLOS ONE compliance.

Add required sections: Data Availability (where the FE model, CLD curves, and raw test data reside), Competing Interests, Author Contributions, Ethics (if any), Funding (already present; check grant numbers). PLOS ONE values replicability; share CLD curves and boundary condition scripts (or a minimal working input deck) in a repository, if possible.

**Minor revisions:**

Title & author block

• Title: “Research on the multi-boundary coupling simulation based on the reaction force of vehicle door’s static sealing system” � tighten and fix possessive.

I suggest changing the title to something like:

“Multi-Boundary Coupled Simulation of Static Sealing Reaction Force in Vehicle Doors.”

Abstract (lines ~15–25)

• Remove generic claims and quantify: replace “errors within 10%” with the exact mean absolute percentage error (MAPE) and n per vehicle.

my suggestion: “Across three vehicle doors (n=3 per model), simulation predicted static closing force within 5-10% MAPE of measurements.”

• Define CLD at first mention.

• Avoid “significant advancement” unless contrasted against a baseline with statistics.

• Final line: state one concrete actionable insight (e.g., “latch-side CLD dominates total force due to lever arm; targeting +X% CLD at latch reduces required closing force by Y%”).

Keywords: Replace “precision” with a domain keyword (e.g., “nonlinear finite elements,” “contact mechanics,” “vehicle sealing,” “NVH”).

I. Introduction (pp. 2–5)

• Focus drift: Early NEV/motor-noise paragraphs are not used in methods/validation; compress to one sentence linking static force to perceived door slam quality. Remove unrelated psychoacoustics unless you leverage them in results.

• Literature gap: Add 1–2 sentences explicitly stating prior door-seal FE works treated the door rigidly or decoupled (moment equilibrium) and did not quantify regional CLD contributions under structural flexibility. Then position your method as addressing (i) door flexibility, (ii) bidirectional interaction, (iii) regional CLD sensitivity.

• Correct typos: “This eformation …” � “This deformation …”. “BRE2” � “RBE2.”

• Claim calibration: Where you say “systematically overlooked,” add a citation or soften to “often neglected.”

The sentence could be revised using something similar to the following.

“We therefore develop a multi-boundary, door–seal coupled model that maps measured/virtual CLD curves to an assembled vehicle FE model, accounts for door flexibility and contact, and quantifies regional contributions to the static closing force; we validate against bench tests on three vehicles.”

II. Numerical Simulation

A. Theoretical introduction (pp. 5–6)

• Definition of static closing force: You define it as the minimum perpendicular force “prior to latch engagement.” Ensure your test follows this (you retracted the latch tongue, good). Keep the definition and test aligned.

• Eq. (1)–(3) clarity:

o Define every symbol immediately under the equation (Fm, Fs, Fx, d0, L, Nx, Lx, Ls).

o Explain F = Fm/100: if Fm is N per 100 mm, say so (e.g., Fm [N/100 mm]); otherwise drop the division.

o In Eq. (3), “E/Ls” appears (scan shows “E / Ls”); confirm it’s not a typo. Present a clean torque balance diagram clarifying the lever arms and sign convention.

o If this subsection serves as the baseline method you supersede, label it “Baseline moment-equilibrium model” and follow with “Proposed coupled FE model.”

B. Sealing system introduction: Ensure Figure 1–3 captions tell the reader what matters (contact interfaces, where closing force is applied, where contributions are evaluated). Label the four “functional zones” on the images.

C. Finite element modeling (pp. 8–11)

• Element types & mesh: Good to note shell midsurfaces. Add whether shells are Kirchhoff or Mindlin, and through-thickness integration points (if relevant to stiffness). Report a mesh convergence check on the closing force (coarse vs. fine).

• Seal representation: If you retain CBUSH springs, explicitly state they implement the nonlinear CLD via a tabular force–deflection curve (include a small table/figure). If you claim hyperelasticity, you must use a solid element seal with a calibrated strain-energy function and surface contact to the door; your current text cites Mooney (1940) and a 2024 strain-energy paper but does not show parameter fitting or testing. Choose one path and describe it fully.

• RBE2/RBE3 usage:

o “Bolts simulated by RBE2” � acceptable, note any pretension (0 if none).

o “Hinges … rotational DOF released using RBE2” � clarify which DOFs are constrained and how the hinge axis is defined.

o Check and correct “BRE2” typos.

• Materials (Table 3): Units look like T/mm3 for density. Consider switching to kg/m3 for readability. Explain the adhesive modulus (1.63 MPa seems very low for a structural adhesive; if it’s a foam/tape, precise it).

• Boundary conditions: Describe body-in-white constraints (are sills/hinge pillars fully fixed?), gravity on/off, and whether glass/regulators are included.

• Workflow (Fig. 6): Add solver branch details (contact, iteration, checks).

D. Segment calibration for CLD (pp. 11–13)

• You mention Method One and Method Two but only say Method One was adopted; describe each method clearly (e.g., virtual cross-sectional compression FE vs. bench compression tests), and show how you fit Mooney–Rivlin/Ogden parameters or, if skipping hyperelastic solids, how you derived the tabular CLD per region (D0−2, D0, D0+2). Define D0 (nominal compression, in mm or %).

II. Analysis Results

A. Simulation results (pp. 13–15)

• Load application: Specify the footprint (pad size) and location at the handle/latch line, and how you detect “full closure prior to latch engagement” in the FE model (target gap = 0? seal compression target?).

• “182 N at the handle”: Provide context; vehicle, configuration, and comparison to test value. Include a reaction-force vs. displacement curve rather than a single value. Report numerical tolerances at convergence.

• Contours (Figs. 10-11): Add color bars with units (N for contact pressure/reaction, if it’s stress, specify MPa; for displacement, mm). Add views that show the four regions.

B. Contribution analysis (pp. 16–18). Clarify vehicle names or keep anonymized consistently (“Sedan A,” “SUV B”).

Conclusion (p. 19)

• Replace “accuracy of 90%” with the measured error statistic and confidence (e.g., “MAPE = 7.1% across 3 vehicles; max error 10%”).

• Add one design takeaway (“raising latch-side CLD by 10% increases total closing force by xx% due to lever-arm dominance, suggesting targeted softening at latch to reduce effort without compromising seal at sill/header”).

References (pp. 20–22)

• Several entries show inconsistent years/volumes (e.g., “International Journal of Mechanical Sciences, vol. 2024, no. 11, 2024”) and malformed DOIs (e.g., SAE). Standardize to PLOS ONE style (authors, year, article title, journal, volume(issue): pages, DOI). Verify each DOI. If you keep the hyperelastic modeling citations, ensure they match your actual implementation (spring vs. continuum).

Reviewers' comments:

Reviewer's Responses to Questions

**Comments to the Author**

1. Is the manuscript technically sound, and do the data support the conclusions?

Reviewer #1: Yes

2. Has the statistical analysis been performed appropriately and rigorously?

Reviewer #1: Yes

3. Have the authors made all data underlying the findings in their manuscript fully available?

Reviewer #1: Yes

4. Is the manuscript presented in an intelligible fashion and written in standard English?

Reviewer #1: Yes

Reviewer #1: Comments

I have completed the review of your manuscript. After review, although your research has some innovative and academic value, the topic selection is relatively novel, the research direction is clear. From theoretical analysis to experimental verification, the overall structure of the article is complete and the data is detailed. Nevertheless, I believe there are still some issues in this manuscript that need further improvement.

1. Your research seems to achieve certain results in simulation and test. However, the title and abstract of the article do not fully reflect the core research content of the article. It is suggested that the title of the article can further highlight the research focus, such as‘the prediction and measurement method of the reaction force of the door sealing system’.

2. The abstract lacks consideration of logical coherence. Please add research background, such as the deficiency in sealing reactive force prediction and breakthrough points in measurement method.

3. The image is not clear enough (such as Fig2), please redraw this Fig and include necessary annotations to ensure clarity and completeness.

4. The lacks consideration of the relationship between Fig3 a b and c; please provide additional commentary on the meaning of the curves in Fig3 c to enhance clarity and understanding.

5. In Section 2.2, the input sources for the sealing strip parameters are not specified. Please provide specific methods and references for obtaining these parameters and explain the rationale and accuracy of the parameter selection.

6. In Section 2.2, please provide a detailed explanation of whether the compression load deflection (CLD) test for the sealing strips was conducted. If the test was performed, please clearly specify the specific methods and steps of the CLD test, including the experimental equipment used, test standards followed, and critical parameters involved. Additionally, please describe the process of sample preparation, including the number, type, and dimensions of the samples, to ensure the scientific rigor and logical consistency of the experimental methods .

7. In Section 2.2, please provide the specific description of simulation method and extraction method of boundary.

8. Section 2.3 please add test preparation, test method, test process and test data extraction, etc.

9. In Section 3.3, The article lacks consideration of the description of test results. Please provide test and experimental validation. Is the verification and comparison between test results and simulation data consistent and representative? Please perform test and experimental validation on 3-5 sets of data and provide detailed explanations of the test results.

10. The description of popularization and application is not sufficient. Please provide the application cases of the method in practical engineering, and discuss the application scope and potential application fields of the method.

11. The conclusion of the thesis is presented in a decentralized manner. Please provide that after the above modification is complete, the conclusion of the thesis be revised, the core achievements and contributions of the research work be summarized, and the value of the research achievements in practical application be analyzed. Based on the research results, the possible future research directions or improvement measures be put forward to make the conclusion of the thesis more clear and complete and to fully show the value and significance of the research.

I look forward to your serious response and improvement to the above questions in the revised draft.

**Do you want your identity to be public for this peer review?** For information about this choice, including consent withdrawal, please see our Privacy Policy

Reviewer #1: No

---

## [Decision Letter · Decision Letter 1]

30 Oct 2025

Prediction Method for the Reaction Force of Vehicle Door Sealing Systems

PONE-D-25-31882R1

Dear Dr. Li,

We’re pleased to inform you that your manuscript has been judged scientifically suitable for publication and will be formally accepted for publication once it meets all outstanding technical requirements.

Kind regards,

Cebastien Joel Guembou Shouop, Ph.D., ME, MS

Academic Editor

PLOS ONE

Additional Editor Comments (optional):

The reviewers’ comments have been adequately addressed, including clarifications on methodology and data presentation.

The revisions have improved the clarity, accuracy, and scientific rigor of the manuscript.

Congratulations on your successful submission.

Reviewers' comments:

Reviewer's Responses to Questions

**Comments to the Author**

Reviewer #1: All comments have been addressed

2. Is the manuscript technically sound, and do the data support the conclusions?

Reviewer #1: Yes

3. Has the statistical analysis been performed appropriately and rigorously?

Reviewer #1: Yes

4. Have the authors made all data underlying the findings in their manuscript fully available?

Reviewer #1: Yes

5. Is the manuscript presented in an intelligible fashion and written in standard English?

Reviewer #1: Yes

Reviewer #1: This study demonstrates that the proposed simulation method for the sealing system reaction force effectively resolves the inaccuracies inherent in simplified Excel-based mathematical models. A key advantage of this method is its incorporation of the coupling effect between the door and the flexible seal, as well as the impact of the seal's post compression deformation on the reaction force. The results have a great influence on the automobile design.

all the questions have been solved and answered, now, the revised paper can be accepted.

**Do you want your identity to be public for this peer review?** For information about this choice, including consent withdrawal, please see our Privacy Policy

Reviewer #1: No

---

## [Editor Report · Acceptance letter]

PONE-D-25-31882R1

PLOS ONE

Dear Dr. Li,

I'm pleased to inform you that your manuscript has been deemed suitable for publication in PLOS ONE. Congratulations! Your manuscript is now being handed over to our production team.

Kind regards,

on behalf of

Dr. Cebastien Joel Guembou Shouop

Academic Editor

PLOS ONE